# Type I interferon is required for T helper (Th) 2 induction by dendritic cells

Lauren M Webb[1,†,§] , Rachel J Lundie[2,‡,§], Jessica G Borger[2], Sheila L Brown[1], Lisa M Connor[3], Adam NR Cartwright[1], Annette M Dougall[4], Ruud HP Wilbers[5], Peter C Cook[1], Lucy H Jackson-Jones[2], Alexander T Phythian-Adams[1] , Cecilia Johansson[6], Daniel M Davis[1], Benjamin G Dewals[4], Franca Ronchese[3] & Andrew S MacDonald[1,*]

## Abstract

Type 2 inflammation is a defining feature of infection with parasitic worms (helminths), as well as being responsible for widespread suffering in allergies. However, the precise mechanisms involved in T helper (Th) 2 polarization by dendritic cells (DCs) are currently unclear. We have identified a previously unrecognized role for type I IFN (IFN-I) in enabling this process. An IFN-I signature was evident in DCs responding to the helminth *Schistosoma mansoni* or the allergen house dust mite (HDM). Further, IFN-I signaling was required for optimal DC phenotypic activation in response to helminth antigen (Ag), and efficient migration to, and localization with, T cells in the draining lymph node (dLN). Importantly, DCs generated from *Ifnar1⁻/⁻* mice were incapable of initiating Th2 responses *in vivo*. These data demonstrate for the first time that the influence of IFN-I is not limited to antiviral or bacterial settings but also has a central role to play in DC initiation of Th2 responses.

**Keywords** dendritic cell; interferon; priming; Th2

**Subject Categories** Immunology; Microbiology, Virology & Host Pathogen Interaction

The EMBO Journal (2017) 36: 2404–2418

## Introduction

Although both helminth infection and allergies exert a devastating global impact and lack effective vaccines or refined therapeutics, basic understanding of the key cell types and mediators that initiate and control type 2 immunity is limited. Despite the potential for innate immune cells such as group 2 innate lymphoid cells to influence type 2 responses (McKenzie *et al*, 2014), it is DCs that are critical for activation and polarization of Th2 immunity (Hammad *et al*, 2010; Phythian-Adams *et al*, 2010; McKenzie *et al*, 2014). However, the signals that DCs provide to facilitate Th2 polarization remain unclear (MacDonald & Maizels, 2008; Bouchery *et al*, 2014).

IFN-I is most well known for its pro-inflammatory role in antiviral immunity (Hoffmann *et al*, 2015). However, this large family of cytokines (including 14 IFNα subtypes in mice, as well as IFNβ) exert diverse effects in a range of infection settings (Bouchery *et al*, 2014; McNab *et al*, 2015). All IFN-I subtypes bind to a common receptor expressed on immune, stromal, and epithelial cells, and studies using IFN-I receptor-deficient mice (*Ifnar1⁻/⁻*) indicate that IFN-I signaling can either enhance or impair the inflammatory response during viral and bacterial infection, depending on context (Ivashkiv & Donlin, 2014; McNab *et al*, 2015). Many IFN-I effects are mediated by a direct impact on DC phenotype and functionality. For example, IFN-I responsiveness controls the ability of CD8α⁺ cDC1s to cross-present viral and tumor Ag to CD8⁺ T cells (Diamond *et al*, 2011; Pinto *et al*, 2011; Ivashkiv & Donlin, 2014) and also influences DC activation, migration, and T cell priming *in vitro* (Parlato *et al*, 2001; Montoya, 2002; Mattei *et al*, 2009; Diamond *et al*, 2011; Pinto *et al*, 2011). Although it has been suggested that IFN-I may be produced by DCs exposed to live *Schistosoma mansoni* eggs (Trottein *et al*, 2004), the role of IFN-I in type 2 inflammation, and how DC function may be modulated by this cytokine family during the orchestration of Th2 responses, is unknown.

We have investigated the key factors involved in DC activation and function during Th2 induction using *in vitro*-generated murine Flt3-L BMDCs (FLDCs), which reflect the heterogeneity and complexity of DC subsets *in vivo* (Naik *et al*, 2005), and using

1 Manchester Collaborative Centre for Inflammation Research, University of Manchester, Manchester, UK
2 Institute of Immunology and Infection Research, Centre for Immunity, Infection and Evolution, University of Edinburgh, Edinburgh, UK
3 Malaghan Institute of Medical Research, Wellington, New Zealand
4 Fundamental and Applied Research in Animals and Health, Immunology-Vaccinology, Faculty of Veterinary Medicine, University of Liege, Liege, Belgium
5 Plant Sciences Department, Laboratory of Nematology, Wageningen University and Research Centre, Wageningen, The Netherlands
6 Respiratory Infection Section, National Heart and Lung Institute, Imperial College London, London, UK
 *Corresponding author. Tel: +44 161 275 1504; E-mail: andrew.macdonald@manchester.ac.uk
 §These authors contributed equally to this work
 †Present address: Baker Institute for Animal Health, Cornell University College of Veterinary Medicine, Ithaca, NY, USA
 ‡Present address: Biomedicine Discovery Institute, Department of Biochemistry and Molecular Biology, Monash University, Clayton, Vic., Australia

*in vivo* models of Th2 priming. These studies indicate that in addition to producing IFN-I in response to helminth Ag and allergens, DC-intrinsic IFN-I signaling is required for their effective migration, localization, and Th2 induction *in vivo*. For the first time, we establish a central role for IFN-I as an early positive regulator of Th2 immunity and demonstrate that IFN-I signaling is essential for optimal DC function during type 2 priming.

# Results

### An IFN-I signature and Th2 induction by FLDCs

To date, studies have failed to identify a defined DC phenotypic activation profile in response to Th2-polarizing helminths (MacDonald & Maizels, 2008; Bouchery *et al*, 2014). Previous work in this area has predominantly been carried out using murine GM-CSF-derived BMDCs (GMDCs) or cell lines (MacDonald *et al*, 2001; Trottein *et al*, 2004). However, FLDCs better represent DC subsets *in vivo* (Naik *et al*, 2005; Helft *et al*, 2015), generating CD24$^{hi}$ equivalents of CD8$\alpha^+$ cDC1s, CD11b$^+$ cDC2s, and plasmacytoid DCs (pDCs; Fig 1A; Brasel *et al*, 2000; Gilliet *et al*, 2002; Guilliams *et al*, 2014). FLDCs were cultured overnight with the potent Th2-inducing soluble egg Ag from *S. mansoni* (SEA; MacDonald & Maizels, 2008), or heat-killed *Salmonella typhimurium* (St) as a Th1/17 control (Perona-Wright *et al*, 2012; Cook *et al*, 2015). Surprisingly, FLDCs secreted significant levels of IFN-I in response to SEA, including IFN$\alpha$3 and IFN$\beta$, in excess of that induced by St (Fig 1B). IFN-I production by FLDCs responding to Th2-Ag was not restricted to SEA, but was also evident following their exposure to the allergen house dust mite (HDM; Fig 1C). Despite significant IFN-I induction, SEA failed to stimulate production of the inflammatory mediators IL-1$\beta$, IL-6, IL-12p40, IL-12p70, or TNF$\alpha$, or regulatory IL-10, from FLDCs (Fig 1D), as reported previously by our group and others using GMDCs or human PBMC DCs (MacDonald *et al*, 2001; de Jong *et al*, 2002). In stark contrast, St induced high-level production of these cytokines (Fig 1D). Although it has recently been suggested that DC-derived IL-10 and IL-33 are important inducers of Th2 responses (Williams *et al*, 2013), we were unable to detect significant levels of either IL-10 (Fig 1D) or IL-33 (Fig EV1A) following FLDC exposure to SEA.

In order to assess whether IFN-I induction by the soluble extract SEA was also evident with whole *S. mansoni* eggs, we cultured FLDCs with live or dead eggs (Fig 1E). This demonstrated significant induction of both IFN$\alpha$3 and IFN$\beta$ following FLDC exposure to dead eggs, as well as a clear IFN-I signature in the form of upregulated IFN-stimulated genes (ISGs), including *Ifit1*, *Mx1*, and *Oas1a* (Fig EV1B). Co-culture of FLDCs with live eggs showed only a trend toward IFN-I secretion and ISG induction, suggesting that the components responsible for DC IFN-I production are predominantly released from dead, or dying, eggs. In addition, IFN-I production was also detected in FLDCs exposed to the T2 ribonuclease (RNase) omega-1 (Everts *et al*, 2009; Wilbers *et al*, 2017), which is the major Th2 immunostimulatory factor in, and one of the most abundant components of, *S. mansoni* egg secretions (Cass *et al*, 2007; Fig 1F).

To identify which FLDC subset(s) produced IFN-I in response to Th2-inducing Ag, cells were sorted prior to SEA stimulation, demonstrating that the primary source was cDC1s (Fig 1G). This was

unexpected, given that pDCs are a key source of this cytokine, particularly in early viral infection (Swiecki & Colonna, 2010). As a complementary approach, we cultured FLDCs generated from mice that express GFP when IFN$\alpha$6 is expressed (Kumagai *et al*, 2007) with SEA, CpG, or HDM (Fig 1H). These IFN$\alpha$6 reporter cells confirmed that in addition to cDC1-restricted production of IFN$\alpha$3 (Fig 1G), both cDC1s and cDC2s had the capacity to produce IFN$\alpha$6 in response to SEA. They also demonstrated that pDCs did not respond to the Th2-associated Ags SEA or HDM with upregulation of IFN-I production and that, although HDM promoted FLDC IFN$\alpha$3 and IFN$\beta$ (Fig 1C), no IFN$\alpha$6 was induced in either cDCs or pDCs by this allergen. As expected, CpG was the most effective IFN$\alpha$6 stimulus in all three FLDC subsets. Together, this shows that cDCs, and particularly cDC1s, can produce IFN-I in response to both helminth Ag and allergens, as has been reported for antiviral settings (Diebold *et al*, 2003; Kato *et al*, 2005).

cDC1s *in vivo* produce IFN-I in response to TLR3-TRIF agonists such as polyI:C (Miyake *et al*, 2009), while pDCs primarily produce IFN-I following TLR7 or TLR9 stimulation in a MyD88-dependent manner (McNab *et al*, 2015). In agreement with this, we found that FLDC production of IFN-I in response to SEA was TRIF-dependent, not MyD88-dependent (Fig 1I). In fact, IFN-I induction was negatively regulated by MyD88 (Fig 1I), a phenomenon reported for TRIF-dependent cytokine responses in macrophages (Johnson *et al*, 2008), but not previously identified as a mechanism of regulating TRIF-dependent IFN-I production in DCs.

In addition to their impact on DC cytokine secretion, SEA and St upregulated expression of surface molecules associated with Ag presentation and costimulation (MHC II, CD40, and CD86) in both cDC1s and cDC2s (Fig EV1C and E), while pDC phenotypic activation was not dramatically altered by either Ag (Fig EV1C). It has recently been suggested that expression of PD-L2 and CD301b may be typical of DCs activated with Th2-inducing Ag (Gao *et al*, 2013; Kumamoto *et al*, 2013). Although upregulation of PD-L2 was observed on cDCs responding to SEA, increased expression of this marker was not restricted to Th2 Ag, but was also evident on DCs responding to St (Fig EV1D and F). Neither Ag influenced the low level of CD301b expressed by any FLDC subset (Fig EV1G). Notably, in most cases, the degree to which activation molecules were upregulated was less striking with SEA than St (Fig EV1). This muted cDC surface activation following exposure to SEA is in keeping with previous reports of GMDCs or human PBMC DCs exposed to SEA *in vitro*, or splenic and hepatic DCs from *S. mansoni*-infected mice, and may reflect an alternative, and more restricted, DC activation phenotype that is common to Th2 settings (MacDonald *et al*, 2001; de Jong *et al*, 2002; Straw *et al*, 2003; MacDonald & Maizels, 2008; Lundie *et al*, 2016).

In order to act as effective Ag-presenting cells (APCs), DCs must be able to migrate to the dLN (Alvarez *et al*, 2008). Since the role of the different DC subsets in Th2 priming is currently unclear (MacDonald & Maizels, 2008; Bouchery *et al*, 2014), and FLDC ability to generate Th2 responses has not yet been addressed, we next assessed their capacity to migrate effectively, and to initiate and polarize immune responses after adoptive transfer into naïve mice. Following injection of dsRed$^+$ FLDCs, cDCs capably trafficked to the dLN (Fig 2A and B). Transferred CD45R$^+$ pDCs could not be detected in the dLN (Fig 2A), in agreement with previous work demonstrating that pDCs can only gain entry to a LN via the blood

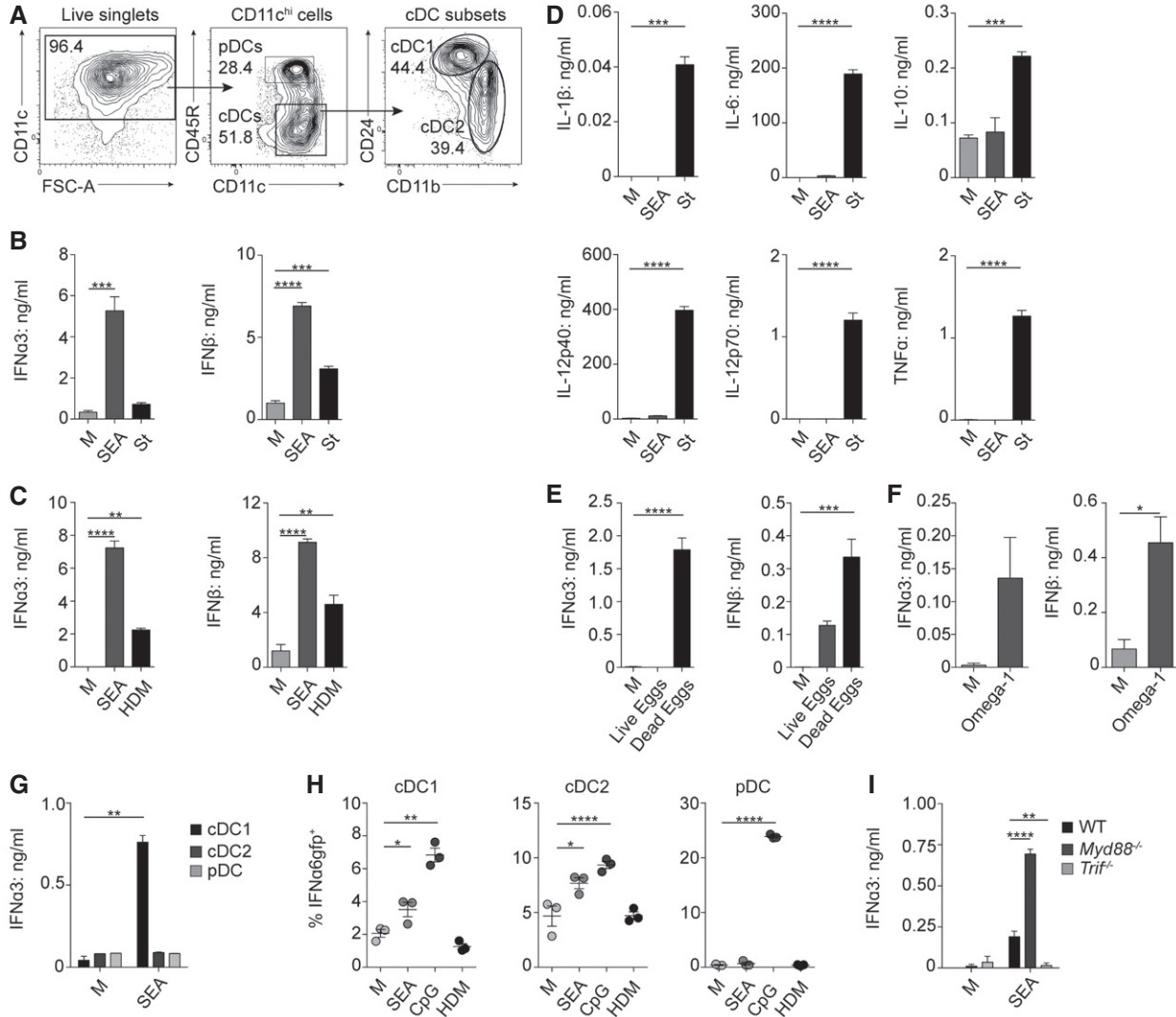

**Figure 1. FLDCs produce IFN-I in response to Th2 Ag.**

A    FLDC subsets were defined by expression of CD11c, CD45R (B220), CD11b, and CD24. pDCs were identified as CD45R⁺ CD11c^lo; cDCs are CD11c⁺ CD45R⁻ with cDC1 and cDC2 subsets.

B, C    IFN-I production as measured by ELISA on supernatants from bulk cultures cultured in medium alone (M) or stimulated with 25 μg/ml soluble egg Ag from *Schistosoma mansoni* (SEA) (B, C), 5 μg/ml *Salmonella typhimurium* (St) (B), or 50 μg/ml house dust mite (HDM) extract (C).

D    Production of DC cytokines from bulk cultures following exposure to SEA or St.

E–G    IFN-I production as measured by ELISA on supernatants from bulk cultures cultured in medium alone (M) or stimulated with 1:500 live or dead whole *S. mansoni* eggs (E), 0.5 μg/ml recombinant omega-1 (F), or FACS-isolated subsets after stimulation with SEA (G).

H    Percentage of IFNα6GFP⁺ DCs after stimulation with SEA, 1 μg/ml CpG, or HDM.

I    IFNα3 production by WT, *Myd88*⁻/⁻, and *Trif*⁻/⁻ bulk FLDCs after exposure to SEA.

Data information: Results are mean ± SEM. *$P < 0.05$, **$P < 0.01$, ***$P < 0.001$, ****$P < 0.0001$ (one-way ANOVA). Data from one of three or more experiments ($n = 3$ replicate wells per group).

through inflamed high endothelial venules and do not routinely migrate via the lymphatics (Diacovo, 2005). While the proportions of cDC1s and cDC2s were approximately 50:50 in cultures prior to transfer (as in Fig 1A), the majority of DCs reaching the dLN were cDC2s (Fig 2A and C), consistent with the suggestion that cDC2s are the primary DC subset that polarizes Th2 responses *in vivo* (Tussiwand *et al*, 2015). Interestingly, even though SEA-exposed cDCs maintained a muted activation phenotype post-transfer (Fig 2D), a characteristic that has also been identified *in vivo* during

*S. mansoni* infection (Straw *et al*, 2003), they competently migrated to the dLN T cell zones (Fig 2E). In addition, despite their limited activation profile, and significant IFN-I production, SEA-exposed FLDCs effectively promoted Th2 polarization following transfer, with clear induction of IL-4, IL-5, IL-10, and IL-13 in dLN restimulations (Fig 2F). As in many Th2 settings (Pearce & MacDonald, 2002), transferred FLDCs also induced SEA-specific IFNγ. It is possible that the minor population of cDC1s present in the dLN were responsible for the low-level IFNγ response that was seen in this

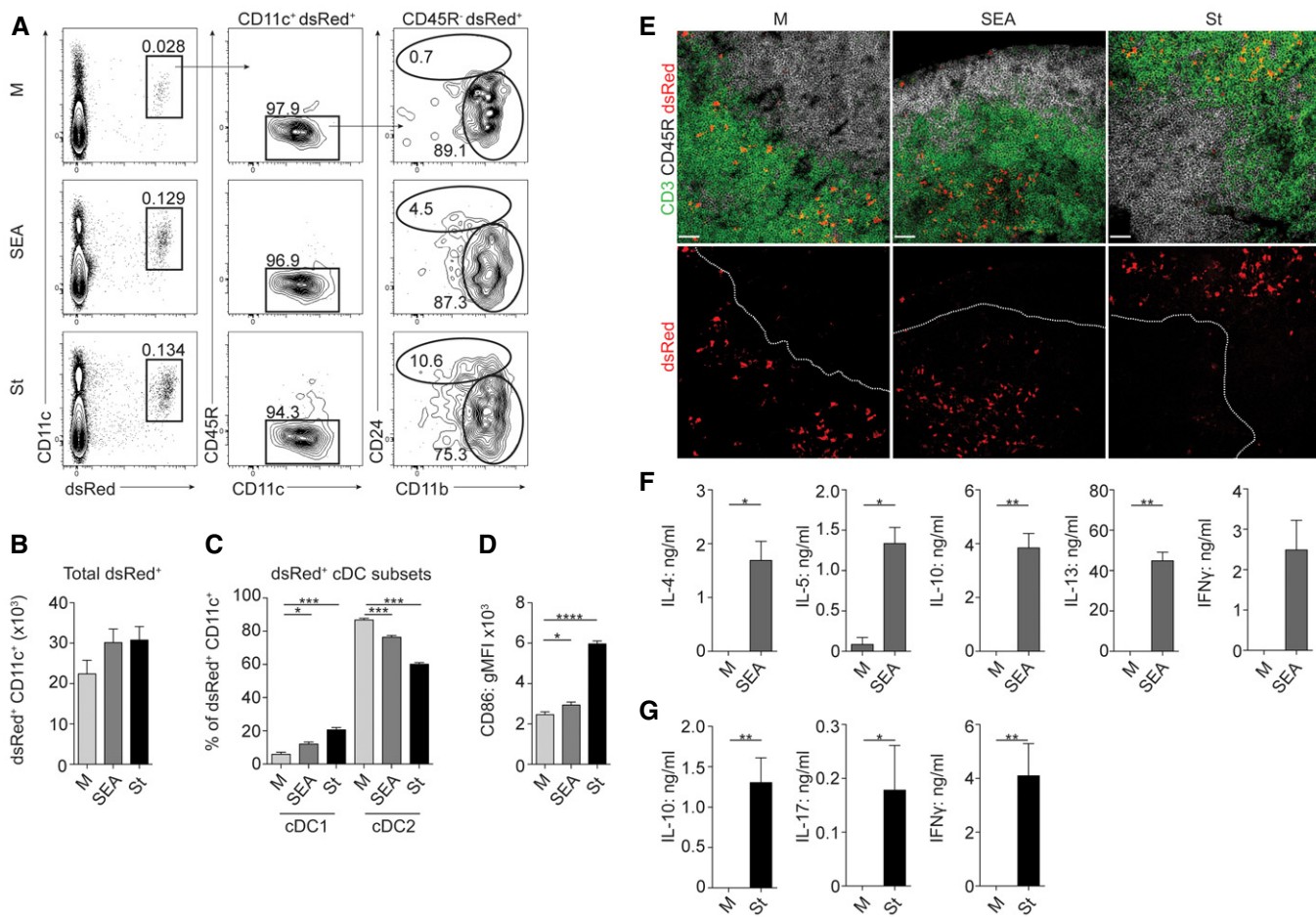

**Figure 2. FLDCs migrate to the dLN and induce Ag-specific responses following adoptive transfer.**

Following Ag stimulation, dsRed⁺ FLDCs were injected s.c. into naïve WT mice, and 48 h later, dLNs were harvested and the presence of dsRed⁺ CD11c⁺ cells was assessed by flow cytometry (A, B) or confocal microscopy (E).

A  dsRed⁺ CD11c⁺ FLDCs were gated as pDCs (CD45R⁺), cDC1s (CD45R⁻ CD24⁺), or cDC2s (CD45R⁻ CD11b⁺).

B  Absolute numbers of dsRed⁺ CD11c⁺ FLDCs in the dLN calculated from flow analysis and cell counts.

C  Percentage of transferred dsRed⁺ CD11c⁺ cells that were cDC1s or cDC2s.

D  CD86 expression on transferred dsRed⁺ CD11c⁺ FLDCs.

E  Confocal microscopy of dLN sections after FLDC transfer: Upper row depicts overlay of CD3 (green), CD45R (gray) and dsRed (red); bottom row depicts dsRed (red) alone. White dashed line represents division between T cell (CD3⁺) and B cell zones (CD45R⁺). Scale bars represent 38 μm.

F, G  Seven days after transfer, dLN cells were restimulated for 72 h with 15 μg/ml SEA (F), 1 μg/ml St (G), or medium alone (M) and cytokine production assessed by ELISA.

Data information: Results are least squares mean ± SEM (B–D) or mean ± SEM (F, G). *P < 0.05, **P < 0.01, ***P < 0.001, ****P < 0.0001 (analyzed using a three-way full-factorial fit model, with contrast analysis used to test differences between experimental groups (B–D) or one-way ANOVA (F, G)). Data from one of three or more experiments (A, E–G) or three experiments pooled (B–D) (n = 3–9 animals per group). gMFI, geometric mean fluorescence intensity.

setting, as has been reported with the *in vivo* equivalents of this subtype (Everts *et al*, 2016). In keeping with their inflammatory phenotype, St-exposed FLDCs capably induced recipient IL-10, IL-17, and IFNγ production (Fig 2G). Consistent with previous reports, no IL-17 was detectable following SEA-activated DC transfer (Larkin *et al*, 2012; Cook *et al*, 2015) and no Th2 cytokines were evident following St-activated DC transfer.

## FLDCs depend on IFN-I signaling for Th2 induction

Having established that SEA triggered FLDC IFN-I secretion (Fig 1), while also conferring Th2-initiation ability (Fig 2), we next addressed the importance of IFN-I production in the Th2 induction

process. Given that DCs themselves can be a target of IFN-I (Montoya, 2002; Mattei *et al*, 2009), we first determined whether IFN-I-responsiveness was required for optimal DC function following SEA stimulation. To this end, we used *Ifnar1⁻/⁻* mice to generate FLDCs lacking the IFNAR1 subunit of the IFN-I receptor, thus rendering them unresponsive to IFN-I (Hwang *et al*, 1995). *Ifnar1⁻/⁻* FLDCs displayed significantly reduced expression of the ISGs *Ifit1* and *Mx1* in response to SEA (Fig 3A), and markedly impaired secretion of IFN-I (Fig 3B), consistent with previous reports using *Ifnar1⁻/⁻* fibroblasts in non-Th2 settings (Marié *et al*, 1998). Additional aspects of SEA-induced activation of WT cDC1s and cDC2s were dramatically impaired in *Ifnar1⁻/⁻* FLDCs, with reduced expression of all markers measured (Fig 3C). IFN-I signaling has been

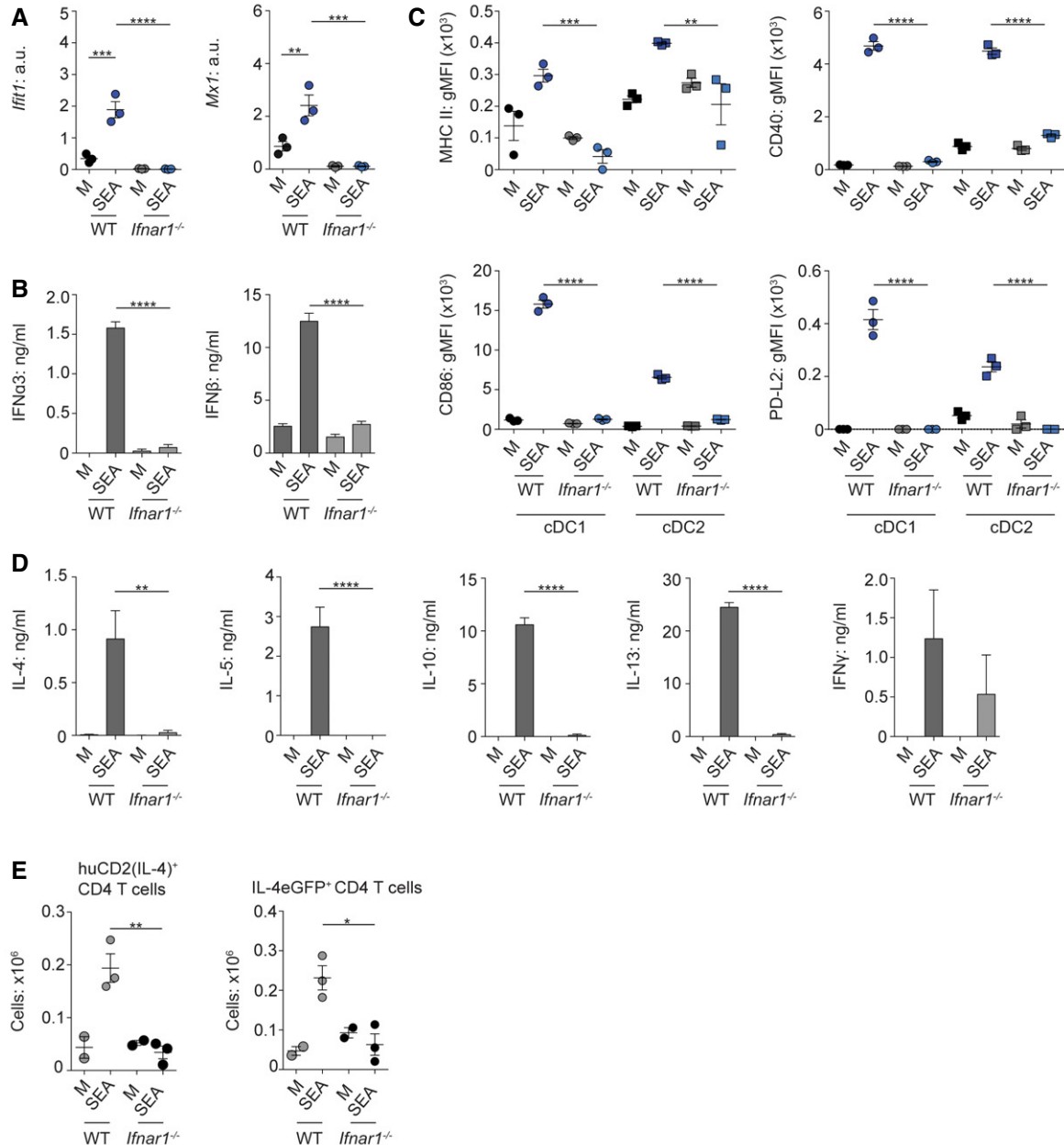

**Figure 3. FLDCs depend on IFN-I signaling for optimal activation and Th2 induction.**

A–D  WT or *Ifnar1*⁻/⁻ FLDCs were cultured with 25 μg/ml SEA or in medium alone (M) for 6 h for gene expression analysis (A) (normalized against *Gapdh*, a.u.) or 18 h for analysis of IFN-I production (B), surface phenotype (C), or injection s.c. into naïve WT mice (D). (D) Seven days after transfer, dLN cells were restimulated for 72 h with 15 μg/ml SEA or medium alone and cytokine production assessed by ELISA (medium-stimulation-alone values subtracted).

E  WT or *Ifnar1*⁻/⁻ FLDCs were transferred s.c. into KN2xIL-4eGFP mice; 7 days later, the presence of IL-4⁺ (huCD2⁺, IL-4 protein) and IL-4eGFP⁺ (IL-4 mRNA) CD4⁺ T cells was assessed by flow cytometry.

Data information: Results are mean ± SEM. *P < 0.05, **P < 0.01, ***P < 0.001, ****P < 0.0001 (one-way ANOVA). Data from one of three or more experiments (n = 3 replicate wells (A–C) or 5 animals (D, E) per group). a.u., arbitrary units. gMFI, geometric mean fluorescence intensity.

implicated in controlling many aspects of DC function, not limited to surface activation and cytokine production, but also regulating their survival and turnover (Mattei *et al*, 2009). However, we did not identify any defect in either differentiation, or survival, of *Ifnar1*⁻/⁻ FL-cDCs (Fig EV2). Most strikingly, in the absence of IFNAR1, the ability of FLDCs to promote SEA-specific Th2 responses

*in vivo* was completely ablated, while IFNγ induction by these DCs was not significantly affected (Fig 3D). Further, using IL-4 reporter mice (Mohrs *et al*, 2005) as recipients, we found that *Ifnar1*⁻/⁻ SEA-activated FLDCs displayed impaired ability to prime CD4⁺ T cell IL-4 secretion (huCD2) or mRNA expression (eGFP) *in vivo* (Fig 3E). Together, our data demonstrate for the first time that IFN-I

responsiveness is essential for optimal DC activation by Th2-inducing Ag and is a key factor governing DC Th2 priming ability *in vivo*.

### cDCs depend on IFN-I signaling for effective migration to, and localization within, the dLN

To determine why $Ifnar1^{-/-}$ FLDCs were impaired in their Th2 induction ability, we first assessed their capacity to process and present Ag to OVA-specific OT-II CD4$^+$ T cells. T cell proliferation was comparable in co-cultures containing WT or $Ifnar1^{-/-}$ cDCs with either OVA protein (OVA) or peptide (pOVA) indicating that $Ifnar1^{-/-}$ cDCs had no defect in their capacity to capture, process, or present Ag *in vitro* (Fig 4A and B). Supporting unimpaired Ag uptake and processing in the absence of IFNAR responsiveness, $Ifnar1^{-/-}$ cDCs internalized and processed DQ-OVA as effectively as WT (Fig 4C). Further demonstrating that $Ifnar1^{-/-}$ cDCs displayed no fundamental deficiency in their ability to polarize CD4$^+$ T cells *in vitro*, both WT and $Ifnar1^{-/-}$ cDCs co-cultured with IL-4, IL-13, or IL-10 reporter CD4$^+$ T cells (Mohrs *et al*, 2005; Kamanaka *et al*,

2006; Neill *et al*, 2010; Cook *et al*, 2015) effectively induced T cell expression of huCD2 (IL-4; Fig 4D), IL-10eGFP (Fig 4E), and IL-13eGFP (Fig 4F) under polarizing conditions. These *in vitro* experiments indicate that DC Ag uptake, processing, and presentation were not significantly impaired in the absence of IFNAR1 signaling.

The ability of $Ifnar1^{-/-}$ cDCs to activate and polarize Th2 cells *in vitro* (Fig 4), but not *in vivo* (Fig 3), strongly suggested that IFN-I responsiveness may be important for effective migration of Th2-promoting DCs. As DC migration through the lymphatic system, and within the dLN, is governed by the chemokine receptor CCR7 and its ligands CCL21 and CCL19 (Braun *et al*, 2011), we next assessed FL-cDC subset expression of CCR7 (Fig 5A). Both cDC subsets expressed low levels of CCR7 on their surface in medium-alone conditions, which may account for the steady-state migration that was evident following adoptive transfer of these unstimulated cells (Fig 2B). cDC1s and cDC2s upregulated CCR7 following SEA exposure, and this was significantly reduced in the absence of IFNAR1 (Fig 5A). However, we saw little expression of CXCR5 on WT or $Ifnar1^{-/-}$ cDCs (Fig 5B), despite effective staining of this chemokine receptor on other cell types, including St-stimulated GMDCs

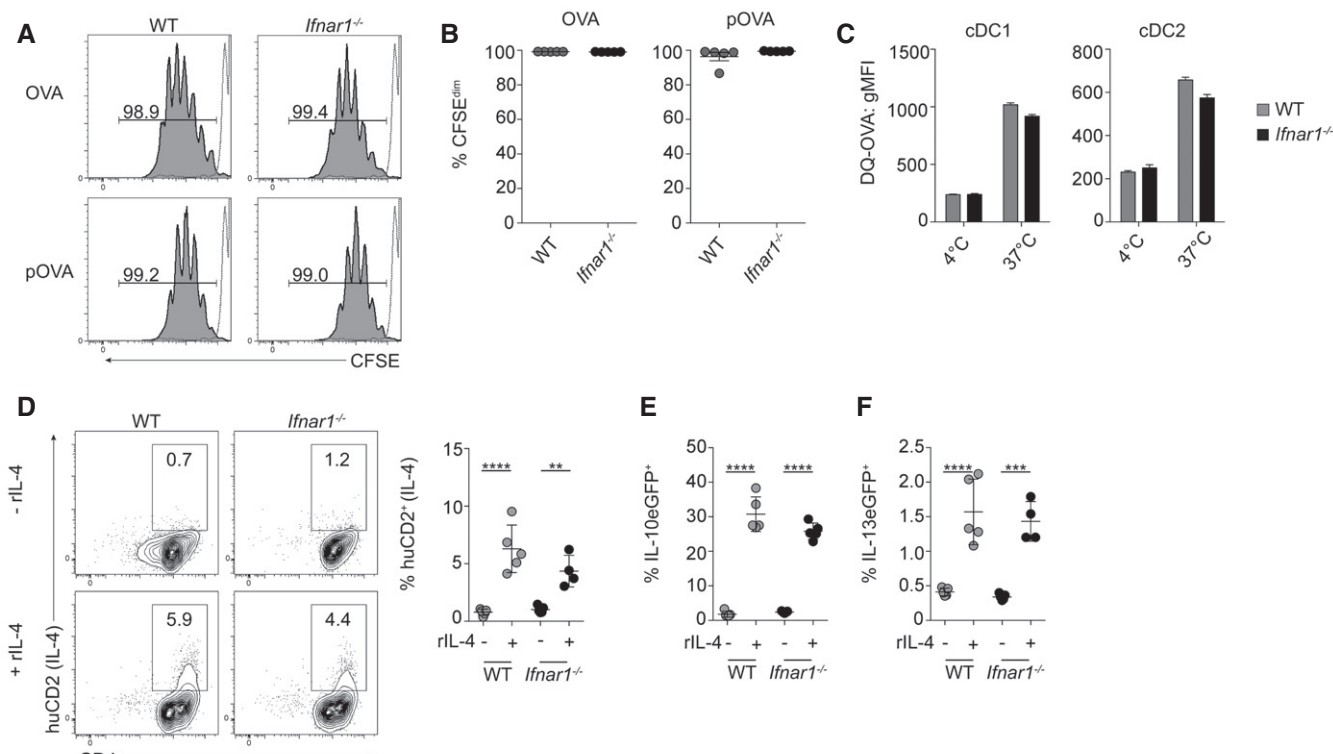

**Figure 4.  $Ifnar1^{-/-}$ FL-cDC APC function is comparable to WT FL-cDC *in vitro*.**

A, B   FL-cDCs were sorted and co-cultured with CFSE-labeled OT-II T cells, with either OVA protein (OVA) or peptide (pOVA), for 96 h. Proliferation was determined by flow cytometric analysis of CFSE dilution. The gray dashed line represents non-proliferating OT-II T cell controls.

C       To assess Ag uptake and processing, WT or $Ifnar1^{-/-}$ FLDCs were cultured with DQ-OVA for 2 h at 37°C or 4°C and uptake assessed by flow cytometry.

D–F   Sorted FL-cDCs were cultured with eGFP$^-$ CD4$^+$ T cells from KN2xIL-13eGFP or KN2xIL-10eGFP animals, with anti-CD3, in the presence (+) or absence (−) of 20 ng/ml rIL-4. IL-4 (huCD2, D), IL-10eGFP (E), or IL-13eGFP (F) expression on CD4$^+$ T cells was assessed by flow cytometry after 72 h of culture. Positive cells were expressed as a percentage of all CD4$^+$ T cells.

Data information: Results are mean ± SEM. **$P < 0.01$, ***$P < 0.001$, ****$P < 0.0001$ (one-way ANOVA). Data from one of three experiments ($n = 4–5$ wells per group).

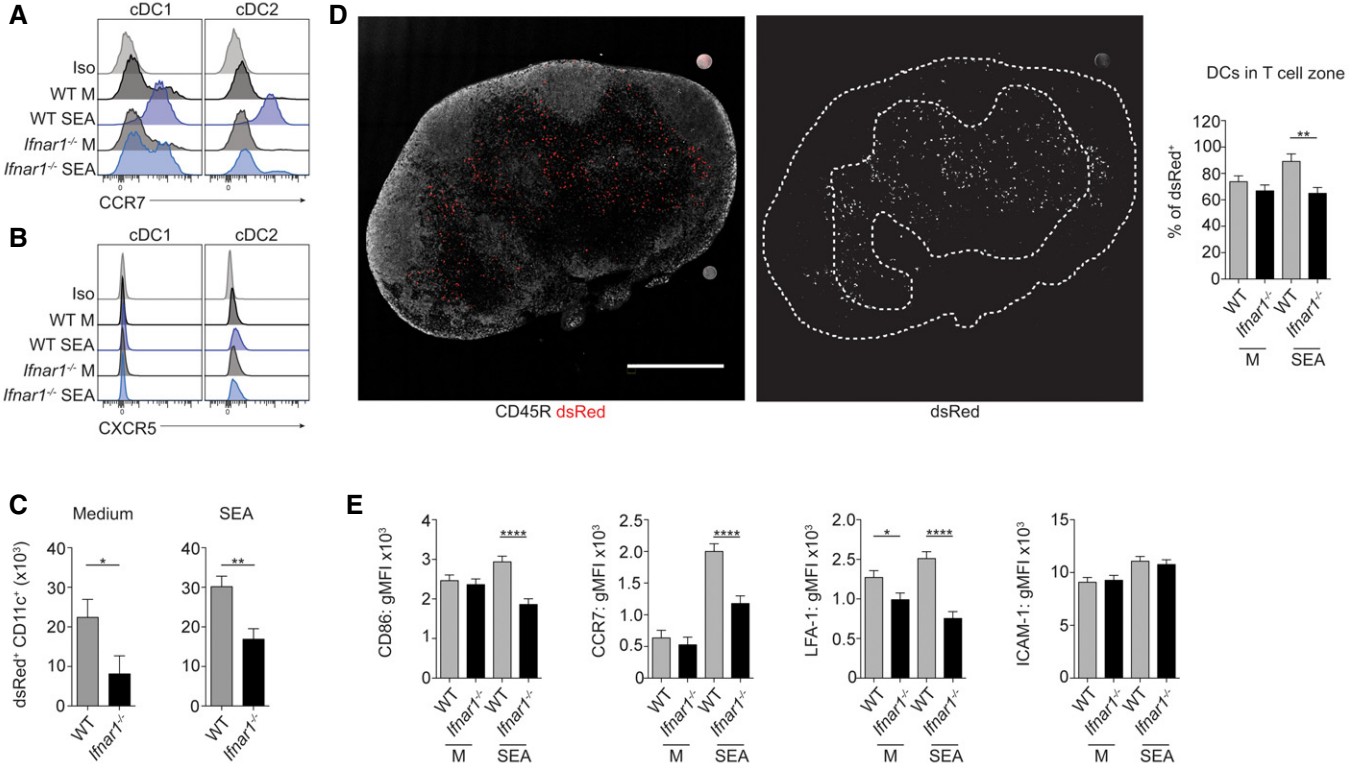

**Figure 5. FL-cDC migration and localization *in vivo* is deficient in the absence of IFNAR signaling.**

A, B  WT or *Ifnar1*$^{-/-}$ FLDCs were cultured for 18 h with 25 µg/ml SEA, or in medium alone (M) and CCR7 (A) and CXCR5 (B) expression assessed by flow cytometry.

C–E  WT or *Ifnar1*$^{-/-}$ dsRed$^+$ FLDCs were injected s.c. into WT mice, and dLNs were harvested 48 h later and analyzed by flow cytometry (C) or confocal microscopy (D) to identify dsRed$^+$ cells. (C) The total number of dsRed$^+$ cDCs in the dLN was calculated from flow data and cell counts. WT data are from the same pooled experiments as Fig 2B. (D) Left image shows an overlay of CD45R$^+$ B cells (gray) and dsRed$^+$ cells (red). Scale bar represents 1 mm. Right image shows dsRed alone (gray), white dashed line represents B cell zone of LN, defined by CD45R staining in left image. Images shown are of dLN following transfer of WT FLDCs cultured in medium alone. The percentage of WT and *Ifnar1*$^{-/-}$ dsRed$^+$ cells within the T cell zone was calculated for each condition. (E) The surface phenotype of transferred WT and *Ifnar1*$^{-/-}$ dsRed$^+$ CD11c$^+$ cells was assessed by flow cytometry.

Data information: Results are least squares mean ± SEM (analyzed using a three-way full-factorial fit model, with contrast analysis used to test differences between experimental groups). *$P < 0.05$, **$P < 0.01$, ****$P < 0.0001$. Data from one of three or more experiments (A, B) or three experiments pooled (C–E) ($n = 3$ wells per group (A, B) or nine animals per group (C–E)).

(Fig EV3A) and T follicular helper cells (Fig EV3B), and in contrast to previous work suggesting that CXCR5 is required for DC induction of Th2 responses (León *et al*, 2012).

Our tracking experiments using WT dsRed$^+$ FLDCs had demonstrated efficient migration of cDCs to dLN T cell zones following adoptive transfer (Fig 2A–C). However, in the absence of IFNAR1 expression, cDC numbers reaching the dLN after injection were reduced by ~50% (Fig 5C). Since CCR7 is required for effective migration of cDCs to, and appropriate localization within, dLN (Braun *et al*, 2011), confocal microscopy was used to determine the location of transferred *Ifnar1*$^{-/-}$ DCs (Fig 5D). While the majority of WT SEA-exposed dsRed$^+$ FLDCs homed to the dLN T cell zones, there was a significant reduction in the proportion of *Ifnar1*$^{-/-}$ SEA-exposed DCs found within this region (Fig 5D). In agreement with our phenotypic studies prior to transfer (Figs 3C and 5A), *Ifnar1*$^{-/-}$ SEA-exposed cDCs expressed lower levels of CD86 and CCR7 on their surface post-transfer, compared to WT (Fig 5E). Similarly, LFA-1 expression was lower on *Ifnar1*$^{-/-}$ DCs compared to WT post-transfer, while ICAM-1 levels were equivalent (Fig 5E).

**IFN-I involvement in Th2 development *in vivo* is not restricted to FLDCs**

Having identified that IFNAR1 plays a key role in regulating activation of FLDCs responding to Th2 Ag *in vitro* (Fig 1), and their ability to initiate Th2 responses *in vivo* (Fig 3), we next assessed the broader role of this signaling pathway in Th2 priming *in vivo*.

We first investigated the expression of ISGs in splenic cDCs purified by FACS from mice challenged intravenously (i.v.) with SEA (Fig 6A). This demonstrated significant upregulation of a range of ISGs by cDCs responding to SEA *in vivo*, including *Ifit1*, *Ifit3*, and *Mx1* (Fig 6B). ISG upregulation was not restricted to the spleen, as mice that had been sensitized intraperitoneally (i.p.) and challenged i.v. with dead *S. mansoni* eggs (Wynn *et al*, 1997) showed significant expression of the ISGs *Mx1* and *Oasl2r* in the lung (Fig 6C). Further, when mice were sensitized intradermally (i.d.) with HDM, cDC expression of the IFN-responsive surface markers CD317 (Blasius *et al*, 2006) and Sca-1 (Essers *et al*, 2009) was significantly upregulated compared to expression

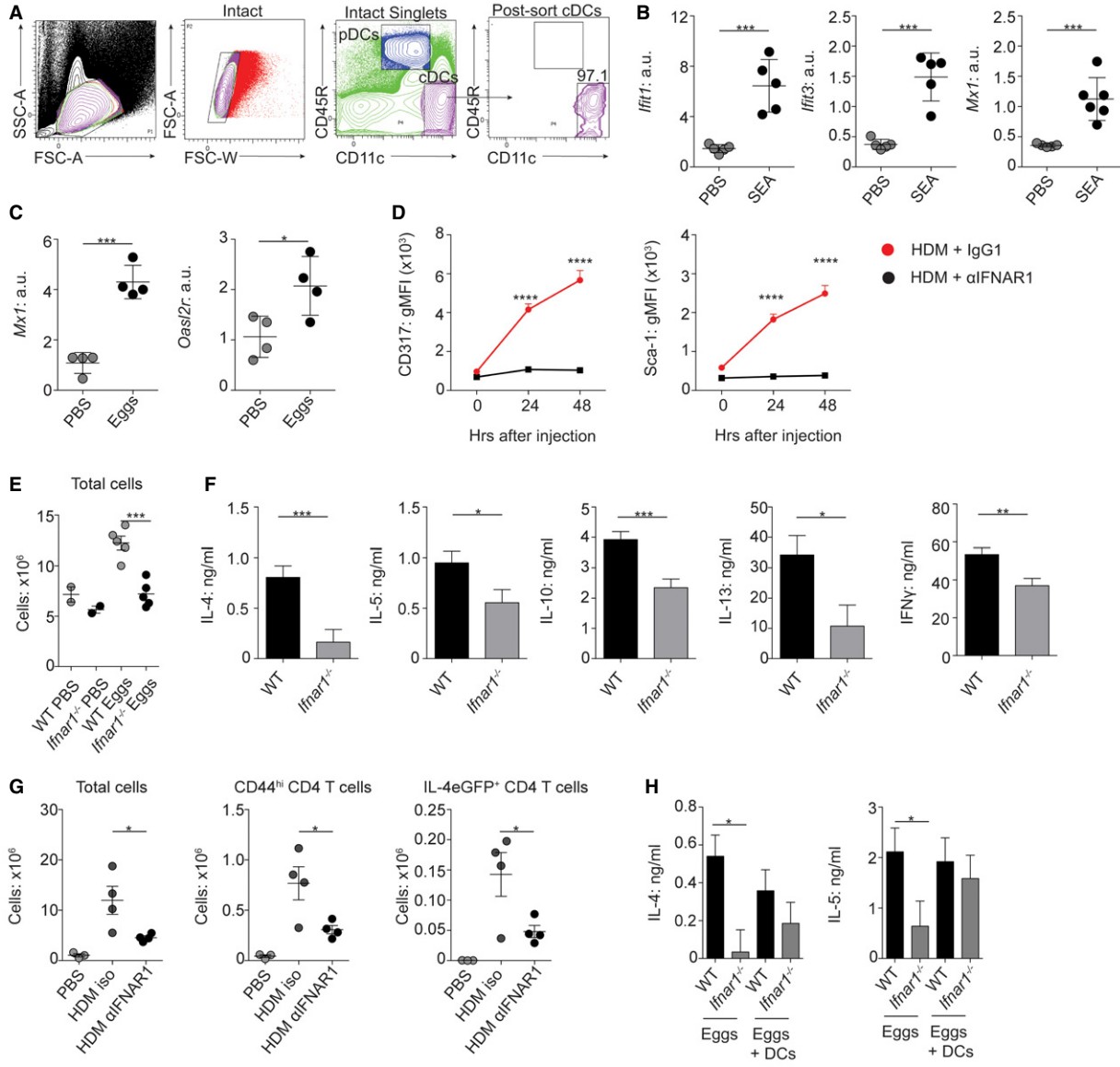

**Figure 6.  IFN-I is required for Th2 induction *in vivo*.**

A   Splenic DCs were FACS-isolated following 12-h exposure to 50 μg SEA administered i.v. Cells were gated as intact singlets and cDCs identified as CD45R⁻ CD11c⁺ cells sorted to > 95% purity.

B   mRNA expression of ISGs by splenic sorted cDCs was assessed by qPCR (normalized against *Gapdh*, a.u.) following *in vivo* exposure to SEA.

C   mRNA expression of ISGs in whole lung following pulmonary challenge with *Schistosoma mansoni* eggs (normalized against *beta-actin*, a.u.).

D   Mice were challenged i.d. with 100 μg HDM in conjunction with 200 μg of the IFNAR1-blocking Ab MAR1-5A3 or an isotype control. Twenty-four or 48 h later, auricular LNs were harvested and CD317 and Sca-1 expression on cDCs analyzed.

E, F   A total of 2,500 *S. mansoni* eggs were injected s.c. per foot into WT or *Ifnar1*⁻/⁻ mice, dLNs were harvested 7 days later, and cell counts were performed (E). (F) Cells were cultured with anti-CD3 for 72 h and then cytokines measured by ELISA (medium background subtracted).

G   IL-4eGFP mice were challenged i.d. with 100 μg HDM in conjunction with 200 μg of the IFNAR1-blocking Ab MAR1-5A3 or an isotype control. Mice were given a second dose of Ab i.p. 48 h later. On day 7, dLNs were harvested and total cell numbers, CD44^hi CD4⁺ T cell numbers, and IL-4eGFP⁺ T cell numbers assessed.

H   A total of 2.5 × 10³ *S. mansoni* eggs were injected s.c. with (+DC) or without 1 × 10⁶ WT FLDCs into WT or *Ifnar1*⁻/⁻ recipients. On day 7, dLNs were harvested, cells were stimulated with anti-CD3 for 72 h, and cytokines were measured by ELISA (medium background subtracted).

Data information: Results are mean ± SEM (B–E, G) (one-way ANOVA) or least squares mean ± SEM (F, H) (analyzed using a three-way full-factorial fit model, with contrast analysis used to test differences between experimental groups). *$P < 0.05$, **$P < 0.01$, ***$P < 0.001$, ****$P < 0.0001$. Data from one of three or more experiments (A–E, G) (*n* = 2–5 animals per group, five replicate wells), or data from three (H) or six (F) experiments pooled. a.u., arbitrary units.

on cDCs from mice treated with the IFNAR1-blocking antibody MAR1-5A3 (Sheehan *et al*, 2006), indicating that expression of these markers was IFN-I-driven (Fig 6D). Together, these data demonstrate that SEA, *S. mansoni* eggs, and HDM stimulate an IFN-I signature *in vivo* and that this is not restricted to FLDCs *in vitro*.

We next assessed the importance of IFN-I responsiveness for Th2 development after direct *in vivo* challenge with either dead *S. mansoni* eggs or HDM. While WT mice displayed significantly increased cellularity in the dLN following *S. mansoni* egg injection, this was not evident in similarly challenged *Ifnar1$^{-/-}$* mice (Fig 6E). Importantly, there was a significant defect in Th2 polarization and IFNγ induction in the dLN of *Ifnar1$^{-/-}$* mice after egg injection (Fig 6F). This was not just a feature of helminth Th2 immunity, or of *Ifnar1$^{-/-}$* mice, as administration of the IFNAR1-neutralizing antibody MAR1-5A3 (Sheehan *et al*, 2006) significantly impaired cellularity, and numbers of dLN CD44$^{hi}$ effector and IL-4-producing CD4$^+$ T cells, in the dLNs of mice sensitized i.d. with HDM (Fig 6G). Finally, this impairment in Th2 induction in *Ifnar1$^{-/-}$* mice after egg injection was "rescued" by co-transfer of WT FLDCs (Fig 6H), indicating that IFN-I responsiveness in DCs alone is sufficient for Th2 initiation in an *Ifnar1$^{-/-}$* environment. Together, these data establish the importance of IFNAR1 signaling in DCs for optimal Th2 polarization against either helminths or allergens *in vivo*.

## Discussion

We have identified IFN-I production by DCs responding to Th2 Ag (Fig 1) and demonstrated for the first time that IFN-I responsiveness is essential for optimal DC activation and effective Th2 response induction *in vivo* (Figs 3 and EV4). This extends our fundamental understanding of the context-dependent role for IFN-I during immune response development, building upon previous studies in which it has been implicated as an important regulator of basal DC activation (Montoya, 2002), and function during viral infection (Pinto *et al*, 2011).

A key aspect of DC activity controlled by IFN-I signaling, which is essential for APC function, is effective migration (Kapsenberg, 2003; Alvarez *et al*, 2008). Studies using *in vitro* migration assays have previously suggested that IFN-I is required for effective transmigration (Mattei *et al*, 2009; Rouzaut *et al*, 2010). Our study is the first to demonstrate the importance of IFN-I in promoting DC migration *in vivo*—from the tissue site, via the lymphatics, to the dLN—in response to pathogen-associated Ag (Fig 5).

Perhaps surprisingly, there is very little work currently published that has focused on the migratory capacity of DCs in a Th2 setting. While it is widely accepted that CCR7 and its ligands CCL19 and CCL21 are the key chemokine receptor–ligand pairs that control DC migration under most circumstances (Alvarez *et al*, 2008), León *et al* (2012) suggested that during helminth infection, the essential chemokine that governs DC Th2 induction is CXCR5 (León *et al*, 2012). CXCR5 interacts with CXCL13 and is required for the formation of germinal centers (Crotty, 2011). CCR7 was expressed to some degree on cDCs cultured in medium alone (Fig 5), perhaps accounting for the homeostatic migration of unstimulated DCs that is evident in Fig 2. Although we identified SEA-specific upregulation of CCR7 expression on cDCs, significant CXCR5 expression was not seen under any circumstances (Fig 5). Further, our data demonstrate an integral role for signaling via IFNAR1 in effective DC Th2 priming and LN T cell zone localization *in vivo*, strongly indicating that DC induction of Th2 responses does not require localization to the fringe of the B cell zone, as has previously been proposed (León *et al*, 2012). It is possible that at later stages of immune activation

during helminth infection, CXCR5 becomes more important for T cell polarization and for T follicular helper cell differentiation (Fairfax *et al*, 2015), while in the early stages of Th2 induction CCR7 plays a more dominant role. Further work is required to determine whether distinct chemokine receptor–ligand pairs are key for DC function at different stages of type 2 differentiation.

As reported by others in non-Th2 settings (Mattei *et al*, 2009), we found that CCR7 upregulation on SEA-activated cDCs was controlled by IFNAR1 signaling, particularly on cDC2s (Fig 5), the primary DC subset responsible for Th2 priming. Additionally, by phenotypic analysis of DCs post-transfer, we identified that optimal expression of the integrin LFA-1 also depended on IFNAR1 (Fig 5). It has previously been shown using human PBMC DCs that LFA-1 facilitates transmigration of DCs across endothelium and is upregulated on DCs by IFN-I treatment (Rouzaut *et al*, 2010). Further, disruption of Rap1, which is required for cell surface stabilization of LFA-1 (Hogg *et al*, 2011), leads to decreased GMDC ability to localize to T cell zones of skin dLN (Katagiri *et al*, 2004). As it has been shown that Rap1 activation is also controlled by IFNAR (Platanias, 2005), this adds further weight to our hypothesis that IFN-I may be influencing cDC migration by facilitating the upregulation of important chemokine receptors and integrins on the surface of cDCs. Thus, it is likely that the combined impact of reduced CCR7 and LFA-1 accounts for defective migration and homing of *Ifnar1$^{-/-}$* FLDCs *in vivo*. In addition, the significant reduction in cDC expression of MHC II, CD40, CD86, and PD-L2 (Fig 3C), all thought to be involved in Th2 induction (MacDonald *et al*, 2001, 2002; Straw *et al*, 2003; Gao *et al*, 2013), likely contributes to the striking inability of *Ifnar1$^{-/-}$* FLDCs to effectively prime Th2 responses *in vivo*. CD40 in particular has been shown to be essential for this process (MacDonald *et al*, 2002), and the significant impairment in CD40 expression is likely to contribute to the phenotype we see *in vivo*.

In our work, despite lower levels of activation (Fig 3), *Ifnar1$^{-/-}$* cDC APC ability was not significantly impaired *in vitro* (Fig 4), contrasting previously published reports that suggested a reduced priming capacity of *Ifnar1$^{-/-}$* cDCs in OVA-specific T cell priming models *in vitro* (Montoya, 2002; Diamond & Farzan, 2013). This discrepancy could be due to technical differences, for example, Ag availability or the ratio of DCs:T cells present, that might alter the importance of cDC IFN-I responsiveness during T cell activation *in vitro*. Conceivably, the strikingly impaired Th2 priming by *Ifnar1$^{-/-}$* DCs that we have identified following their adoptive transfer into recipient mice (Fig 3) may in part be due to restricted Ag availability and a relatively small Ag-specific T cell pool *in vivo* in WT mice.

An important aspect of DC function in type 2 settings that remains poorly understood is the pattern recognition receptors (PRRs) and pathways used by DCs to respond to Th2-associated Ag, and how these might influence Th2 priming. Here, we demonstrate that the FLDC IFN-I response to SEA is dependent on the adaptor protein TRIF, while MyD88 acts as a negative regulator of this pathway (Figs 1 and EV4). Identification of TRIF as an important adaptor protein for DC activation by SEA suggests a role for the TRIF-dependent PRRs in sensing IFN-I inducing ligands following exposure to Th2 Ag (Fig EV4). Such PRRs are primarily nucleic acid sensing and include TLR3, TLR4, and the cytoplasmic receptor complex DDX1–DDX21–DDX36 (Broz & Monack, 2013; McNab *et al*, 2015). The role of MyD88 as a negative regulator of PRR signaling

has not previously been reported in a Th2 setting, although it is known that Th2 induction is not impaired in the absence of splenic DC, GMDC, or global MyD88 expression (Layland *et al*, 2005; Jankovic *et al*, 2002, 2004; Marshall & Pearce, 2008). Our data demonstrate a requirement for TRIF in optimal FLDC activation in response to a Th2 Ag and highlight the need for further study in this area to address the impact of TRIF deficiency on DC Th2 induction. Additionally, our demonstration of FLDC IFN-I induction by the key immunostimulatory molecule in SEA omega-1 (Fig 1) suggests that glycoproteins and C-type lectin receptors such as the mannose receptor, which has been shown to recognize omega-1 (Everts *et al*, 2012), may be involved in this process.

Having identified that there was a degree of differential expression of the IFN-I subtypes IFNα3, IFNα6, and IFNβ by cDCs responding to Th2 Ag (Fig 1), it would be of interest to characterize cDC expression of additional IFN-I subtypes in response to different Th2 Ags. Whether distinct subtypes are more or less responsible for conferring effective Th2 induction ability on cDCs remains to be shown, although current understanding is that the diverse IFN-I subtypes exert similar biological function in viral infection (van Pesch *et al*, 2004).

The question of *why* IFN-I induction, normally associated with intracellular infections, has evolved against a helminth parasite, and whether this IFN-I signal benefits the host and/or the parasite, remains. IFN-I has previously been suggested to act as a damage signal that enhances early innate immune activation (Gregorio *et al*, 2010). Many of the PRRs that induce IFN-I production sense nucleic acid ligands (McNab *et al*, 2015). Our work comparing the IFN-I signature induced by live or dead eggs (Figs 1 and EV1) suggests that the molecules responsible for IFN-I induction are primarily components released by dead or dying eggs. This would include omega-1, as well as implicating ligands such as nucleic acids that are not normally secreted by live intact eggs. In addition, omega-1 is a RNase that degrades messenger RNAs (Everts *et al*, 2012), likely leading to the production of self-nucleic acid ligands. This RNase activity is essential for the Th2-polarizing capacity of omega-1 (Everts *et al*, 2012). Thus, it is possible that one way that Th2 Ag enhances polarization of Th2 cells may be via the stimulation of IFN-I production, a previously unrecognized role for this damage-associated signal. It has also been shown that IFN-I can limit IL-12 production upon secondary stimulation with TLRs (McNab *et al*, 2015); thus, IFN-I may also promote Th2 polarization by blocking differentiation down the Th1 route.

Our identification of this central role for IFN-I in DC Th2 promotion raises many new questions about the broader impact of IFN-I during type 2 inflammation. Importantly, the DC focus of our work does not exclude additional roles for IFN-I in the wider Th2 setting, and how IFN-I may influence effector functions of other immune cells during helminth infection remains to be determined. Similarly, our identification of a DC-derived source of IFN-I does not rule out its production by additional cell types exposed to helminth products or allergens *in vivo* (Fig EV4). Discovery of a key role for IFN-I in DC Th2 ability has implications not only for development of therapeutics aimed at modulating type 2 responses, but also in the context of helminth co-infection with bacterial or viral pathogens, where IFN-I is known to impact dramatically on disease outcome (McNab *et al*, 2015). In future work, it will be important to define whether IFN-I induced in response to helminth Ag is a long-term cost, or benefit, to pathogen and host.

Together, our work emphasizes the pleiotropic nature of IFN-I and the importance of considering the previously unappreciated role of this family of cytokines, ordinarily associated with antiviral immunity, in promotion of type 2 inflammation.

# Materials and Methods

### Mice

C57BL/6xIL-4eGFP, dsRed, dsRedx*Ifnar1*$^{-/-}$, *Ifnar1*$^{-/-}$, *Myd88*$^{-/-}$, IFNα6GFP, KN2xIL-4eGFP, KN2xIL-10eGFP, KN2xIL-13eGFP, OT-IIxLy5.1, and *Trif*$^{-/-}$ mice, all on the C57BL/6 background, were bred and maintained at the University of Edinburgh, the University of Manchester, or the Malaghan Institute of Medical Research, under specific pathogen-free conditions (Hwang *et al*, 1995; Adachi *et al*, 1998; Barnden *et al*, 1998; Yamamoto, 2003; Vintersten *et al*, 2004; Mohrs *et al*, 2005; Kamanaka *et al*, 2006; Kumagai *et al*, 2007; Neill *et al*, 2010). Control C57BL/6 (WT) mice were either bred and maintained in-house or obtained commercially from Harlan or the Jackson Laboratory. Experiments were conducted under a license granted by the Home Office (UK), in accordance with local guidelines, or under approval from the Victoria University Animal Ethics Committee, and performed according to Malaghan Institute guidelines (HDM challenge in IL-4eGFP mice), or under approval from the ethics committee of the University of Liege (*S. mansoni* egg lung challenge model).

### Cell culture

For FLDC generation, BM cells were cultured with 200 ng/ml Flt3-L (PeproTech), according to published methods (Naik *et al*, 2010). Briefly, BM cells were flushed from femurs and tibia of adult mice and resuspended for counting after red blood cell (RBC) lysis. Cells were then cultured at 37°C 5% $CO_2$ at $1.5 × 10^6$ cells/ml in RPMI 1640 medium (Sigma) containing Flt3-L, 10% fetal calf serum (Sigma), 2 mM L-glutamine, 50 mM 2-mercaptoethanol (Invitrogen), 50 U/ml penicillin, and 50 μg/ml streptomycin (Sigma). FLDCs were harvested on d8 and replated at $2 × 10^6$/ml for 18 h in the presence or absence of Ag. For gene expression analyses, cells were cultured for 6 h in medium alone or with Ag. Cells were exposed to 25 μg/ml SEA, 5 μg/ml St, 50 μg/ml HDM (*Dermatophagoides pteronyssinus*; Greer Laboratories), 1:500 live *S. mansoni* eggs (purified from the livers of C57BL/6 mice at approximately day 49 of *S. mansoni* infection), or dead eggs (freeze/thawed) or 500 ng/ml recombinant omega-1 (generated in *Nicotiana benthamiana* and purified from the leaf extracellular space using POROS 50 cation resin (Life Technologies) (Wilbers *et al*, 2017). Endotoxin-free SEA from *S. mansoni* eggs was prepared in-house as previously described (MacDonald *et al*, 2001). In brief, *S. mansoni* eggs were homogenized using a Tenbroeck tissue grinder in PBS, and supernatant was collected, centrifuged, filtered (0.45 μm), and stored at −70°C. St, strain SL3261, was provided by Dr. Maurice Gallagher (University of Edinburgh) and was heat-killed by incubation at 80°C for 30 min and stored in PBS at 4°C until use.

### FLDC sorting and flow cytometry

For some experiments, FLDC bulk cultures were sorted by FACS into separate subset populations prior to Ag stimulation. FLDCs

were harvested on d8 of culture, and following FcR-Block (2.4G2), cells were surface-stained with mAbs against CD11c, CD45R (B220), CD11b, and CD24. Sorted or unsorted DC populations were replated at $2 \times 10^6$/ml and stimulated with Ag for 18 h as described above. Following Ag stimulation, DC surface activation was assessed by flow cytometry. Cells were first stained with LIVE/DEAD Fixable Aqua or UV (Life Technologies) and, following FcR-Block, were stained with combinations of the following mAbs: CCR7, CD11b, CD11c, CD24, CD40, CD45R, CD301b, CXCR5, CD86, MHC II, and PD-L2. For analysis of cell viability, cells were stained with Annexin V and 7-AAD. All antibodies for flow cytometry were purchased from BD, eBioscience, or BioLegend. IFNα6GFP FLDCs were also stained with the above antibodies and their expression of GFP analyzed by flow cytometry. To assess antigen uptake, $2 \times 10^5$ FLDCs were incubated with 10 µg DQ-OVA (Life Technologies) for 2 h at 37°C or 4°C, before flow cytometric analysis. Samples were acquired on a FACSCanto II or LSR-Fortessa flow cytometer using BD FACSDiva Software and analyzed with FlowJo software (Tree Star, Inc.) to ascertain proportional expression and geometric mean fluorescence intensity (gMFI) for surface marker expression.

### Cell tracking and transfer

For cell tracking and transfer experiments, WT, dsRed$^+$, Ifnar1$^{-/-}$, or Ifnar1$^{-/-}$ dsRed$^+$ FLDCs were cultured as above for 6 or 18 h, with SEA or St or in medium alone. For tracking of dsRed$^+$ cells, FLDCs were injected subcutaneously (s.c.) into WT recipients ($1 \times 10^6$/foot). Following dsRed$^+$ cell transfer, dLNs were harvested at 48 h post-injection. Popliteal LNs (pLNs) were digested at 37°C for 30 min with 0.4 U/ml Liberase TL (Roche) and 80 U/ml DNase I type IV (Sigma). Single-cell suspensions were prepared by mechanical disruption of LNs through a 70-µm filter. LN suspensions were then analyzed by flow cytometry for the presence of dsRed$^+$ CD11c$^+$ cells. dLN cells were first stained with LIVE/DEAD Fixable Aqua or UV, and following FcR-Block, cells were stained with mAb against CD11b, CD11c, CD24, CD45R, CD86, CCR7, ICAM-1, and LFA-1. Identification of transferred cells was performed following exclusion of Lin$^+$ cells using CD3, CD19, CD49b, Ly6G, and NK1.1. Monocytes (Ly6C$^+$ CD11b$^+$) and macrophages (CD11c$^-$ F4/80$^+$) were also excluded. For cell transfer experiments, FLDCs were injected into WT recipients ($0.25 \times 10^6$/foot) and dLNs harvested 7 days later. Single-cell suspensions of LN cells ($5 \times 10^6$ cells/ml) were cultured in X-VIVO 15 medium (Lonza) containing 2 mM L-glutamine and 50 µM 2-ME (Life Technologies) in 96-well plates at 37°C 5% CO$_2$ and stimulated with SEA (15 µg/ml) or St (1 µg/ml). Supernatants were harvested after 72 h and cytokine production assessed by ELISA.

### Immunohistochemistry

Following dsRed$^+$ cell transfer, dLNs from recipient mice were fixed in 4% paraformaldehyde (PFA; Sigma), incubated sequentially in 15 and 30% sucrose (Sigma) solutions overnight to quench residual PFA, and then mounted in Optimal Cutting Temperature (O.C.T.) embedding medium (Sakura). Thin-section confocal microscopy was performed on 20-µm cryostat sections, which were fixed in ice-cold acetone for 10 min, and then incubated for 1 h with mAb against CD3 conjugated to 488 (eBioscience, UK), and CD45R conjugated to Alexa647 (BD Biosciences, UK). Slides were washed extensively in

PBS–Tween-20, PBS, and then water before nuclei were stained with DAPI-supplemented ProLong Fade Gold (Life Technologies) mounting media. Samples were imaged at room temperature (~23°C) using TCS SP5 II or TCS SP8 CW microscopes (Leica Microsystems), with laser lines with emission wavelengths of 405 (DAPI), 488 (Alexa Fluor 488), 543 (dsRed), and 633 and 647 (Alexa Fluor 647) nm, with 20× (HC PL APO; N.A. = 0.75), 40× (HCX PL APO CS; N.A. = 1.25), and 63× (HCX PL APO; N.A. = 1.4) objectives with immersion oil (Type F, Leica), using LAS AP or LAS X software (Leica). Fluorophore emission light was collected using PMT or Hybrid Detectors (HyD). Data were rendered and analyzed using Volocity (Improvision) or ImageJ (NIH) software. To analyze the localization of dsRed$^+$ cells in specific dLN regions, the total LN area was measured on a binary image of the DAPI staining, the B cell zone area identified using CD45R staining, and the T cell zone as the CD45R$^-$ region. A threshold of 65-255 (the mean intensity of background in uninjected samples) was applied to the dsRed channel and the total area of dsRed ascertained. dsRed$^+$ cells within the T cell zone were then calculated as a percentage of all dsRed$^+$ cells within the LN.

### Immunizations and *in vivo* treatments

To assess the IFN-I signature of *in vivo* DCs responding to Th2 Ag, mice were injected intravenously with PBS or 50 µg SEA. Eighteen hours later, spleens were digested (as pLN, above) for 15 min at 37°C, single-cell suspensions prepared by mechanical disruption through 70-µm filters and low-density cells enriched using NycoPrep (1.077 g/ml; Axis-Shield). For purification of splenic DCs, non-DC lineage cells were removed using biotinylated mAbs against murine CD2, CD3ε, CD49b, mIgM, and erythrocytes (Ter-119) and MyOne Streptavidin Dynabeads (Invitrogen). Cells were then stained with antibodies against CD11c and CD45R and cDCs sorted as CD11c$^{hi}$ CD45R$^-$ directly into RNALater (Ambion) for subsequent RNA extraction. To assess Th2 priming *in vivo*, $2.5 \times 10^3$ dead *S. mansoni* eggs were injected s.c. per foot into recipient mice. In some experiments, $1 \times 10^6$ WT FLDCs were co-transferred with *S. mansoni* eggs. Draining pLNs were harvested 7 days later, and restimulated with plate-bound αCD3 (0.5 µg/well) for 72 h. An alternative model of *in vivo* priming involved intradermal (i.d.) challenge into the ear pinnae with 100 µg HDM whole bodies. Auricular dLNs were collected 24 h, 48 h, or 7 days later. Flow cytometry was used to analyze CD45R$^-$ cDC expression of CD317 and Sca-1 at 24 h or 48 h, and to assess the number of IL-4eGFP$^+$ T cells at day 7. In these experiments, mice were either treated with an anti-IFNAR1 antibody (MAR1-5A3) or an isotype control (MOPC-21; both from BioXcell), with 200 µg Ab mixed with HDM given i.d. Mice received a second Ab dose 48 h later i.p. In order to address induction of ISG expression following dead *S. mansoni* egg challenge of the lung, mice were injected with 5,000 dead *S. mansoni* eggs i.p. On day 14 mice were challenged with 5,000 dead eggs i.v. Lungs were then collected on day 22 and a portion reserved for gene expression analysis.

### ELISA

ELISAs were performed on culture supernatants using paired mAb and recombinant cytokine standards, or DuoSets (BioLegend, eBioscience, BD, R&D Systems, and PeproTech). For restimulation data,

medium-alone values were subtracted from Ag restimulated cytokine levels for each sample.

### RNA extraction and qPCR analysis

RNA was recovered from FLDCs or splenic DCs using TRIzol reagent (Life Technologies) or RNALater, respectively. Lung tissue was prepared using the mirVana miRNA isolation kit (Life Technologies). RNA was translated into cDNA using Superscript III Reverse Transcriptase and Oligo (dT; Life Technologies). Quantitative RT–PCR was performed using a LightCycler 480 II Real-Time PCR machine (Roche) and LightCycler-DNA master SYBR Green I (Roche) and compared to a serially diluted standard of pooled cDNA. The relative amount of mRNA for genes of interest was normalized to GAPDH or beta-actin. The following primers were used: *β-actin*: 5′-AGAGGGAAATCG TGCGTGAC-3′, 5′-ACGGCCAGGTCATCACTATTG-3′; *Gapdh*: 5′-AAT GTGTCCGTCGTGGATCT-3′, 5′-CCCAGCTCTCCCCATACATA-3′; *Ifit1*: 5′-TCTAAACAGGGCCTTGCAG-3′, 5′-GCAGAGCCCTTTTTGATAATG T-3′; *Ifit3*: 5′-TGAACTGCTCAGCCCACA-3′, 5′-TCCCGGTTGACCTCAC TC-3′; *Mx1*: 5′-TTCAAGGATCACTCATACTTCAGC-3′, 5′-GGGAGGTG AGCTCCTCAGT-3′; *Oas1a*: 5′-GCTGCCAGCCTTTGATGT-3′, 5′-TGGC ATAGATTGTGGGATCA-3′; and *Oasl2r*: 5′-GGATGCCTGGGAGAGAA TCG-3′, 5′-CAGTTTCGAAGAGCAGGCGA-3′.

### DC: T cell co-culture assays

For CFSE dilution assays, splenic and LN CD4$^+$ OT-II T cells were purified using CD4$^+$ Dynabeads (Life Technologies) following the manufacturer's protocol. T cells were labeled with 5 μM CFSE (Life Technologies) for 15 min at 37°C, prior to culture with $5 \times 10^4$ WT or *Ifnar1$^{-/-}$* FLDCs in the presence of 0.01 μg/ml OVA$^{323-339}$ peptide (CRB) or 5 μg/ml OVA protein (Sigma) which had been endotoxin depleted in-house. Cultures were incubated at 37°C for 4 days prior to assessment of CFSE dilution by flow cytometry. *In vitro* co-culture polarization experiments were performed with $5 \times 10^4$ KN2xIL-13eGFP$^-$ CD4$^+$ or KN2xIL-10eGFP$^-$ CD4$^+$ T cells which were cultured in 96-well plates for 4 days with 2,500 WT or *Ifnar1$^{-/-}$* FLDCs, 1 μg/ml soluble anti-CD3, and with or without rIL-4 (20 ng/ml; PeproTech; Cook *et al*, 2015). T cell expression of huCD2 (IL-4; Life Technologies) or eGFP (IL-10 or IL-13) was then assessed by flow cytometry.

### Statistical analysis

Statistical analyses were carried out using GraphPad Prism 5 or JMP (SAS Institute). The data were checked to confirm normality and that groups had equal variance. One-way analysis of variance (ANOVA) with Tukey's multiple comparison tests was employed to determine significant differences between sample groups. Results from these tests were reported as significant if $P \leq 0.05$, with results from these tests shown as mean ± SEM. For some experiments, statistical analysis was carried out using JMP, in which case data were analyzed using three-way full-factorial fit models to assess effects such as "genotype", "treatment", and "experiment" on the response variable of interest. This allowed the interaction between effects to be taken into account in addition to their impact on the response variable, which enabled experimental repeats to be pooled, increasing the power of the analysis. The least squares mean results table from the three-way full-factorial analysis was used to test the contrast between specific experimental groups using a joint *F*-test. A difference between experimental groups was taken to be significant if the *P*-value (Prob > *F*) was less than or equal to 0.05, with results in graphs shown as least squares mean ± SEM.

**Expanded View** for this article is available online.

### Acknowledgements

This work was supported by the MRC (G0701437 to A.S.M.) and the Wellcome Trust (WT086628MA). R.J.L. is the recipient of a National Health and Medical Research Council of Australia (NHMRC) Early Career Fellowship. Funding was also provided to F.R. by the Health Research Council of New Zealand. B.G.D. is a senior researcher from the F.R.S-FNRS. A.M.D. was funded by the ULg-Marie Curie COFUND Program. C.J. is supported by the MRC (G0800311). The authors thank John Grainger, James Hewitson, and Mark Travis for critical reading of the manuscript, Rinku Rajan for technical support, Richard Preziosi for statistical advice, Martin Waterfall and Gareth Howell for cell sorting and assistance with flow cytometry, and David Gray, Caetano Reis e Sousa, Andrew McKenzie, Markus Mohrs, and Shizuo Akira for provision of OT-II × Ly5.1, *Myd88$^{-/-}$*, *Trif$^{-/-}$*, IL-10eGFP, *Ifnar1$^{-/-}$*, IL-13eGFP, KN2, and IFNα6GFP mice. *Biomphalaria glabrata* snails used to generate eggs and SEA for this research were supplied by the N.I.A.I.D. Schistosomiasis Resource Center at the Biomedical Research Institute (Rockville, MD), through Contract HHSN272201000005I for distribution through BEI Resources, with the help of Mark Wilson.

### Author contributions

LMW and RJL designed the project and coordinated and carried out the experimental work. JGB and ANRC helped design the project and carried out the microscopy work. SLB, AMD, LMC, LHJ-J, PCC, and ATP-A carried out the experimental work. PCC, LHJ-J, and ATP-A contributed to the design of *in vitro* and *in vivo* assays. RHPW contributed recombinant omega-1. DMD provided some equipment and supervised some of the research. CJ provided IFNα6GFP mice. BGD and FR supervised some of the research. ASM conceived and designed the project and supervised the research. LMW, RJL, and ASM wrote the manuscript, with valuable input from all the other authors.

### Conflict of interest

F.R., A.S.M., and L.M.C. are listed as inventors on a provisional patent application concerned with the subject matter of this paper. The MCCIR is a joint venture between the University of Manchester, GSK, and AstraZeneca.

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
