## [Review Process File · The EMBO Journal]

Manuscript EMBO-2016-95345

A central role for Type I IFN in Th2 response induction by dendritic cells

Lauren M Webb, Rachel J Lundie, Jess G Borger, Sheila L Brown, Lisa M Connor, Adam NR Cartwright, Annette M Dougall, Ruud HP Wilbers, Peter C Cook, Lucy H Jackson-Jones, Alexander T Pythian-Adams, Cecilia Johansson, Daniel M Davis, Benjamin G Dewals, Franca Ronchese, Andrew S MacDonald

Corresponding author: Andrew MacDonald, University of Manchester

Review timeline:

Submission date:	26 July 2016
Editorial Decision:	29 October 2016
Revision received:	08 March 2017
Accepted:	16 May 2017

Editor: Bernd Pulverer

Transaction Report:

1st Editorial Decision

29 October 2016

Thank you for submitting your manuscript for consideration by the EMBO Journal and both for your patience in awaiting this delayed formal decision and the constructive pre-decision discussion. The three referee reports have been appended again below and we have studied your proposed revision plan in some detail.

I would like to invite you to submit a revised version of the manuscript, addressing the comments of all three reviewers largely along the lines you have suggested in your response dated 25th October. Below, we clarify a number of points arising from your response:

ref 1:

- 1) +ve control for CXCR5: we agree this is valuable addition and would request this.
- 2) 'rescue the *Ifnar1* deficit with DC transfers': as discussed, we believe this is an excellent suggestion and we would encourage you to address this experimentally. In this context, it is worth adding that referee 2 commented back to us on reading the reports of the other two referees 'It is true that a detailed analysis of *in vivo* DCs of *IFNAr1/fl x CD11cCre* mice would be great and valuable'. The referee also reaffirms his interest in the dataset in this message and issues this comment as a suggestion, rather than a 'must have'. We appreciate this is an involved experiment, but two of our referee have clearly emphasized that the *in vivo* relevance could be developed and we agree with this assessment. The experiment you suggested would appear to address this request (if it is conclusive): 'rescue the impaired Th2 response in the *IFNaR* KO mice by transferring in WT FLDCs along with the schistosome eggs.'
- 3) conditional ko mice: we will not require this.

4) challenge HDM animals via skin and/or lungs: this is a useful suggestion, but it will not be a requirement for publication in EMBO Journal.

ref 2:

- 1) 'nomenclature for DC subsets': We need to defer to you and the expert referees on this issue. Please suggest your preferred response and we will discuss it further. Of course, this issue shall not affect the publication of this manuscript.
- 2) 'Why would only dead cells and not live eggs trigger the IFN response; release of TLR3 ligands?': It would be useful if there was experimental evidence that cytokines are released.
- 3) 'DCs from HDM exposed animal': this would be useful data and we would gladly include it in the paper, but it will not be a precondition for publication.

ref 3:

- 1) 'Is Omega-1 the major functional component of HDM?': this is a useful suggestion, but it will not be a requirement for publication in EMBO Journal.
- 2) 'Which IFN-I, IFN-a3 or IFN-a6, primarily contributes to IFN-I signaling?': please do add this data in revision.
- 3) 'why were total numbers of dsRed+ CD11c+ DCs in DLN of SEA- and St- treated mice similar with that of control mice?': please address this issue textually.
- 4) 'test the underlying mechanisms that IFN-I signaling induced activation and CCR7 upregulation by DCs': will not be a requirement for publication in EMBO Journal.
- 5) add schematic: we agree this would be valuable and if required our artist can assist.

I should add that it is EMBO Journal policy to allow only a single round of revision, and acceptance of your manuscript will therefore depend on the completeness of your responses in this revised version.

When preparing your letter of response to the referees' comments, please bear in mind that this will form part of the Review Process File, and will therefore be available online to the community. For more details on our Transparent Editorial Process, please visit our website: http://emboj.embopress.org/about#Transparent_Process

Thank you for the opportunity to consider your work for publication. I look forward to your revision.

REFEREE REPORTS

Referee #1:

The authors analyze the ability of in vitro cultured dendritic cells to stimulate Th2 cell responses. Instead of using GM-CSF-stimulated DCs, the authors use Flt3-cultured DCs (FLDCs) that purportedly have features more similar to in vivo DC subsets. The authors stimulated these cells with schistosome antigens (SEA) and look for cytokine and cell surface receptor expression. They identified that certain type 1 interferons are upregulated in FLDCs following SEA stimulation, and these cells can stimulate T cells to produce Th2 cytokines in vitro and in vivo. Thus the type 1 interferons are stimulating both Th1 and Th2-like profiles. Transfer of FLDCs generated from *Ifnar1* KO mice failed to induce Th2 cell responses to schistosome eggs. The deficit was not in APC function, but in the migration of the *Ifnar1*-deficient FLDCs.

Major issues

A significant issue with this paper is that the vast majority of the author's data relate to in vitro-derived FLDC cells. As such the link to the in vivo situation is potentially tenuous. Although the

authors show limited data from Ifnar-KO mice in Fig 6, the experiments do not specifically link the effect to DCs.

The authors comment on the absence of CXCR5. It would be helpful to have a positive control since this staining can be difficult.

Could the authors attempt to rescue the Ifnar1 deficit with DC transfers?

It would be more convincing if the authors could use Ifnar1 fl/fl mice with an appropriate Cre strain to delete Ifnar1 in DCs. These mice are available from Jax.

This paper is predominantly cellular immunology and would be better placed in a more specialised immunology journal.

Minor issues

Page 7 line 155 should read (Supplementary Fig 1c and d)

Referee #2:

In this paper, the author make the discovery that type I inteferon signalling intrinsically in DCs is involved in Th2 cell priming to schistosoma egg antigen, to Omega1 of Schistosoma, and to HDM allergen. The authors use Flt3L cultured DCs to make this point, and show that mainly CD11b+ cDC2s are drivers of type 2 immunity in response to IFNAR signalling. They also show the TRIF pathway of signalling to be involved, while myd88 inhibits this pathway. There are very few papers dealing with the cytokine signals that promote Th2 immunity, and therefore this paper is highly novel. The only backdrop would be that a lot of experiments are done with in vitro generated DCs.

I have only a few remarks

1. There is a new nomenclature for DC subsets, that the authors might consider using. CD8 pos or equivalent CD24hi DCs are now called cDC1, while CD11b+ DCs are now called cDC2
2. Why would only dead cells and not live eggs trigger the IFN response. Any release of TLR3 ligands ?
- 3; Any evidence for a type I interferon signature in DCs directly sorted out of a HDM exposed animal ?
4. The authors should also refer to older literature where others investigated the role of IFN and IFNAR in OVA-models of asthma. The literature might be quite fragmented.
5. The HDM model employed is just a primary immune response induction. What about challenge of these animals via the skin or lungs. Does lack of IFNAR reduce features of atopic dermatitis or asthma ? These experiments might be beyond the scope of the current paper, but nonetheless important for future research.

Referee #3:

In this manuscript the authors investigated the role of type I IFN in the helminths-induced Th2 response. Interestingly, they firstly discovered that helminths stimulation induced the production of IFN-I by DCs. Moreover, they identified that IFN-I signaling is required for the activation and draining lymph node-homing of DCs, which lead to the activation of T cells in draining lymph node. The data are clearly presented and the results are convincing. However, there are a few clarifications that could improve the work.

1. In Results, line 118-120 "...omega-1, which is the major Th2 immunostimulatory component of SEA", as known, the major functional component of HDM are immunogenic epitopes, including der p or der f families during allergic asthma. Is Omega-1 the major functional component of HDM to induce the IFN-I production by DCs?
2. In figure 1g and h, the author demonstrated that CD24hi DCs produce IFN-a3 and IFN-a6, while CD11bhi DCs only produce IFN-a6 under SEA stimulation, and both DC subsets were significantly activated by SEA stimulation in IFN-I signaling dependent manner. Which IFN-I, IFN-a3 or IFN-a6, primarily contributes to IFN-I signaling?
3. Fig. 2b: As the SEA and St stimulation could activate and induce DLN-homing of DCs, why the

total numbers of dsRed+ CD11c+ DCs in DLN of SEA- and St- treated mice were similar with that of control mice?

4. The authors found that the major DC population migrated to DLN after SEA stimulation was CD11bhi DCs but not CD24hi DCs in figure 2b. However, SEA could upregulate the CCR7 expression on both CD24hi and CD11bhi DCs, is there any difference in CD24hi and CD11bhi DC intercellular signaling which was induced by IFN α R activation? Did the authors test the underlying mechanisms that IFN-I signaling induced activation and CCR7 upregulation by DCs?

5. It is a little confusing about the kinetics of the IFN-I production by DCs and the activation process of Th2 responses. It seemed that SEA activated IFN-I production by DCs, then IFN-I acted on DCs themselves via IFN α R and upregulated the expression of CCR7 which induced migration of DC to DLNs. It would help to better understand this process if the authors can provide a schematic figure.

1st Revision - authors' response

08 March 2017

Referee #1:

The authors analyze the ability of in vitro cultured dendritic cells to stimulate Th2 cell responses. Instead of using GM-CSF-stimulated DCs, the authors use Flt3-cultured DCs (FLDCs) that purportedly have features more similar to in vivo DC subsets. The authors stimulated these cells with schistosome antigens (SEA) and look for cytokine and cell surface receptor expression. They identified that certain type 1 interferons are upregulated in FLDCs following SEA stimulation, and these cells can stimulate T cells to produce Th2 cytokines in vitro and in vivo. Thus the type 1 interferons are stimulating both Th1 and Th2-like profiles. Transfer of FLDCs generated from Ifnar1 KO mice failed to induce Th2 cell responses to schistosome eggs. The deficit was not in APC function, but in the migration of the Ifnar1-deficient FLDCs.

Major issues

A significant issue with this paper is that the vast majority of the author's data relate to in vitro-derived FLDC cells. As such the link to the in vivo situation is potentially tenuous. Although the authors show limited data from Ifnar-KO mice in Fig 6, the experiments do not specifically link the effect to DCs.

Response:

We would highlight that FLDCs are widely accepted to represent DC subsets *in vivo*, generating equivalents of CD8 α + cDCs (cDC1s), CD11b+ cDCs (cDC2s) and plasmacytoid DCs. We and others have used *in vitro* generated DCs extensively to discover key new facets of their activation and function (e.g. see Cook et al., Nature Communications 2015; Rigby et al., EMBO 2014), information that has in many cases been instrumental in leading to complementary discoveries *in vivo*. Our use of these cells in the current study has enabled – for the first time – the refined characterization of truly naïve DC subset responses to helminth and allergen challenge, along with adoptive transfer experiments to allow tracking studies and unequivocal restriction of *Ifnar1* deficiency to the priming DCs. We would suggest that this use of adoptive transfer of DCs provides one of the most direct links to the *in vivo* situation that could possibly be achieved with current technology. Such definitive experiments would be very difficult to achieve with the very limited numbers of DCs (of uncertain activation status) that could be obtained via alternative approaches, such as sorting *ex vivo* from digested tissues.

To support our FLDC data, in Figure 6 of the original manuscript we demonstrated *in vivo* induction of an IFN-I signature 1) in splenic DCs following i.v. injection of SEA and, 2) in the lung in response to i.v. *S. mansoni* egg challenge. Additionally, we showed that mice globally deficient in *Ifnar1* expression displayed a markedly impaired Th2 response following direct s.c. challenge with schistosome eggs *in vivo* and that Ab-mediated blockade of IFNAR1 significantly reduced Th2 development *in vivo* following intradermal administration of HDM allergen.

We have added substantial new data to the revised manuscript to specifically address the

reviewer's concern and further support the *in vivo* relevance of our work. We have now shown that IFN-I dependent markers are upregulated on the surface of DCs *in vivo* in response to HDM treatment (Figure 6d of the revised manuscript). Perhaps most importantly, we have also shown that transfer of WT FLDCs 'rescues' the induction of Th2 cytokines in *Ifnar1*^{-/-} mice following the injection of *S. mansoni* eggs (Figure 6h of the revised manuscript).

Together, we believe that the novel data we have generated using these *in vivo* models supports the broad relevance of our detailed FLDC work, providing a platform for future work beyond the scope of the current study more directly linking DCs, IFN-I and Th2 polarization during active Th2 infection or allergic inflammation *in vivo*. Please also see comments on advantages and disadvantages of CD11c restricted transgenic models, below.

The authors comment on the absence of CXCR5. It would be helpful to have a positive control since this staining can be difficult.

Response:

We have added this data to Supp. Figure 3 of the revised manuscript, where we show significant induction of CXCR5 on the surface of BMDCs in response to *S. typhimurium* (St) and on activated T follicular helper cells in the hepatic LN during *S. mansoni* infection.

*Could the authors attempt to rescue the *Ifnar1* deficit with DC transfers?*

Response:

We thank the reviewer for this constructive suggestion. Although DC reconstitution is technically difficult, we have spent considerable time and effort optimising this approach to be able to add data to the revised manuscript that more directly links the FLDC and *Ifnar1*^{-/-} recipient data in our original submission. As described above, our novel data shows that WT DCs can 'rescue' the deficit in egg-specific Th2 induction evident in *Ifnar1*^{-/-} mice (Figure 6h), indicating that IFN-I responsiveness in DCs alone is sufficient for Th2 initiation in an *Ifnar1*^{-/-} environment.

*It would be more convincing if the authors could use *Ifnar1* fl/fl mice with an appropriate *Cre* strain to delete *Ifnar1* in DCs. These mice are available from Jax.*

Response:

While we agree that use of the approach suggested by the reviewer would complement the data presented in our study, we do not think they would alter the core message, that of discovery of a novel role for IFN-I and *Ifnar1* in DC activation by helminths and allergens, and DC ability to induce Th2 responses following their transfer *in vivo*. Unfortunately, experiments using transgenic mice with CD11c-restricted disruption of *Ifnar1* were not possible for our study, as this would have required establishment, rederivation, expansion and maintenance of complex colonies that is beyond our current resources. Further, we would respectfully remind the reviewer that adoptive transfer of DCs is a refined and accepted method to test fundamental aspects of their functional ability *in vivo* in a system that unequivocally restricts the deficiency of interest to the transferred DCs. In contrast, there is currently no system available that definitively targets DCs alone *in vivo*, with transgenic models reliant on CD11c restriction impacting a range of other cell types in addition to DCs. The widespread expression of *Ifnar1* in the immune system, and of CD11c by (some) macrophage populations, is likely to limit the clarity and DC specificity of results that would be generated by such an approach.

This paper is predominantly cellular immunology and would be better placed in a more specialised immunology journal.

Response:

We would suggest that the core discoveries of our study will be of interest to a wide readership, not restricted to immunologists, relating as they do to novel and fundamental mechanisms involved in coordination of inflammation and immunity against both helminths

and allergens.

Minor issues

Page 7 line 155 should read (Supplementary Fig 1c and d)

Response:

Apologies for this error, which we have corrected.

Referee #2:

In this paper, the author make the discovery that type I inteferon signalling intrinsically in DCs is involved in Th2 cell priming to schistosoma egg antigen, to Omega1 of Schistosoma, and to HDM allergen. The authors use Flt3L cultured DCs to make this point, and show that mainly CD11b+ cDC2s are drivers of type 2 immunity in response to IFNAR signalling. They also show the TRIF pathway of signalling to be involved, while myd88 inhibits this pathway. There are very few papers dealing with the cytokine signals that promote Th2 immunity, and therefore this paper is highly novel. The only backdrop would be that a lot of experiments are done with in vitro generated DCs.

Response:

We thank Reviewer 2 for these positive comments. Please see response to Reviewer 1 on the topic of FLDCs and the expanded *in vivo* readouts in our revised submission.

I have only a few remarks

1. There is a new nomenclature for DC subsets, that the authors might consider using. CD8 pos or equivalent CD24hi DCs are now called cDC1, while CD11b+ DCs are now called cDC2.

Response:

Following on from this helpful comment, we have altered the paper to use this nomenclature, which we hope helps with clarity.

2. Why would only dead cells and not live eggs trigger the IFN response. Any release of TLR3 ligands ?

Response:

This is an interesting point that we discussed in the original manuscript to some extent, and have expanded upon in the discussion in our revised manuscript, to reflect the addition of data to Supplementary Figure 1 that demonstrates that ISGs are significantly induced in FLDCs in response to dead eggs, but only marginally in response to live egg exposure. This data indicates that the main components that drive the IFN-I response are released from dying eggs, and may also relate to comparatively lower concentrations of nucleic acids or other ligands that are likely released from live vs. dead eggs.

3; Any evidence for a type I interferon signature in DCs directly sorted out of a HDM exposed animal ?

Response:

Acting upon this interesting suggestion, we have now shown that the IFN-I dependent markers CD317 and Sca-1 are upregulated on the surface of DCs in response to HDM *in vivo*, in an IFNAR1 dependent manner (new Figure 6d of the revised manuscript).

4. The authors should also refer to older literature where others investigated the role of IFN and IFNAR in OVA-models of asthma. The literature might be quite fragmented.

Response:

Apologies, but we are not sure which literature the reviewer is referring to here. We are unaware of any published link between IFN1, IFNAR and asthma.

5. The HDM model employed is just a primary immune response induction. What about

challenge of these animals via the skin or lungs. Does lack of IFNAR reduce features of atopic dermatitis or asthma? These experiments might be beyond the scope of the current paper, but nonetheless important for future research.

Response:

We agree that the suggested HDM experiments would be very interesting, but beyond the scope of the current manuscript to address experimentally in a timely fashion.

Referee #3:

In this manuscript the authors investigated the role of type I IFN in the helminths-induced Th2 response. Interestingly, they firstly discovered that helminths stimulation induced the production of IFN-I by DCs. Moreover, they identified that IFN-I signaling is required for the activation and draining lymph node-homing of DCs, which lead to the activation of T cells in draining lymph node. The data are clearly presented and the results are convincing. However, there are a few clarifications that could improve the work.

Response:

We thank Reviewer 3 for these positive comments.

1. In Results, line 118-120 "...omega-1, which is the major Th2 immunostimulatory component of SEA", as known, the major functional component of HDM are immunogenic epitopes, including der p or der f families during allergic asthma. Is Omega-1 the major functional component of HDM to induce the IFN-I production by DCs?

Response:

Omega-1 is a component specific to SEA, and not found in HDM. We would suggest that identification of the specific HDM components that are responsible for DC IFN-I production, while an interesting future direction, is beyond the scope of the current manuscript.

2. In figure 1g and h, the author demonstrated that CD24hi DCs produce IFN-a3 and IFNa6, while CD11bhi DCs only produce IFN-a6 under SEA stimulation, and both DC subsets were significantly activated by SEA stimulation in IFN-I signaling dependent manner. Which IFN-I, IFN-a3 or IFN-a6, primarily contributes to IFN-I signaling?

Response:

While it is known the IFN-I subsets have different levels of specificity for IFNAR, it is not currently known which IFN-I is the most potent in eliciting responses following binding, particularly in the Type 2 setting. We have added discussion to the revised manuscript to highlight this interesting point, and the potential for distinct IFN-I subtypes to play different roles in the Th2 induction process.

3. Fig. 2b: As the SEA and St stimulation could activate and induce DLN-homing of DCs, why the total numbers of dsRed+ CD11c+ DCs in DLN of SEA- and St- treated mice were similar with that of control mice?

Response:

We would expect a basal level of steady-state migration by adoptively transferred DCs to the draining lymph nodes, particularly as the control (media alone) DCs express a low level of CCR7 (see Figure 5a, relative to isotype staining). We have added text to the revised manuscript to clarify this point.

4. The authors found that the major DC population migrated to DLN after SEA stimulation was CD11bhi DCs but not CD24hi DCs in figure 2b. However, SEA could upregulate the CCR7 expression on both CD24hi and CD11bhi DCs, is there any difference in CD24hi and CD11bhi DC intercellular signaling which was induced by IFNaR activation? Did the authors test the underlying mechanisms that IFN-I signaling induced activation and CCR7 upregulation by DCs?

Response:

Beyond demonstrating the importance of IFNAR responsiveness for CCR7 upregulation by DCs responding to SEA (Figure 5a) we have not yet interrogated the signalling events involved in this process.

5. It is a little confusing about the kinetics of the IFN-I production by DCs and the activation process of Th2 responses. It seemed that SEA activated IFN-I production by DCs, then IFN-I acted on DCs themselves via IFNAR and upregulated the expression of CCR7 which induced migration of DC to DLNs. It would help to better understand this process if the authors can provide a schematic figure.

Response:

We thank the reviewer for this constructive suggestion, are happy to provide a schematic, and have added a draft of such to the revised manuscript for the reviewers and editors to assess (Supplementary Figure 4). We hope that this schematic aids with clarity and in conveying the key points of our manuscript.

Accepted

16 May 2017

Thank you for your continued patience in awaiting the second decision. We recently received the report of referee #2 and we will forego the report of referee #1 at this point. As you will see below, ref #2 enthusiastically supports publication. In our view the key issues of ref #1 have been successfully addressed where appropriate - we agreed before that the *Ifnar1* fl/fl mouse experiment suggested is not essential. We have reviewed the author checklist and consider all essential points to be adequately represented.

I am therefore very pleased to inform you that your manuscript has been accepted for publication in the EMBO Journal. My colleagues will be in touch with any additional requests and forms to ensure a speedy publication at this stage to make up for some of the delays.

If you have any questions, please do not hesitate to call or email the Editorial Office. Thank you for your contribution to The EMBO Journal.

REFEREE REPORT

Referee #2:

The authors have addressed all my concerns. I was already very positive in the first round of review, and this has only gotten better. The authors say there are no murine studies on type I IFN in the OVA/alum model. They could refer to : Nakajima,.... Iwamoto. J Immunol 1994 Where it is shown that type I IFN suppresses OVA induced eosinophilic airway inflammation

PS: Please note that it is EMBO Journal policy for the transcript of the editorial process (containing referee reports and your response letter) to be published as an online supplement to each paper. If you do NOT want this, you will need to inform the Editorial Office via email immediately. More information is available here: http://emboj.embopress.org/about#Transparent_Process

Your manuscript will be processed for publication in the journal by EMBO Press. Manuscripts in The EMBO Journal will be copy edited, and you will be provided with page proofs prior to publication. Please note that any supplementary information is not included in the proofs.

Corresponding Author Name: Prof. Andrew MacDonald

Journal Submitted to: EMBOJ

Manuscript Number: EMBOJ-2016-95345R